# Time-Series Metabolome and Transcriptome Analyses Reveal the Genetic Basis of Vanillin Biosynthesis in Vanilla

**DOI:** 10.3390/plants14131922

**Published:** 2025-06-23

**Authors:** Zeyu Dong, Shaoguan Zhao, Yizhang Xing, Fan Su, Fei Xu, Lei Fang, Zhiyuan Zhang, Qingyun Zhao, Fenglin Gu

**Affiliations:** 1Spice and Beverage Research Institute, Chinese Academy of Tropical Agricultural Sciences, Hainan Key Laboratory of Genetic Improvement and Quality Control of Tropical Spice and Beverage Crops, Key Laboratory of Processing Suitability and Quality Control of the Special Tropical Crops of Hainan Province, National Center of Important Tropical Crops Engineering and Technology Research, Wanning 571533, China; 2Hainan Institute, Zhejiang University, Sanya 572025, China; 3Zhejiang Provincial Key Laboratory of Crop Genetic Resources, Key Lab of Plant Factory for Plant Factory Generation-Adding Breeding of Ministry of Agriculture and Rural Affairs, The Advanced Seed Institute, Zhejiang University, Hangzhou 310058, China; 4Sanya Research Institute, Chinese Academy of Tropical Agricultural Sciences, Hainan Key Laboratory for Biosafety Monitoring and Molecular Breeding in Off-Season Reproduction Regions, Sanya 572025, China

**Keywords:** *Vanilla planifolia* Andrews, vanillin, combined analysis, transcriptome and metabolome

## Abstract

Vanillin, the principal aromatic compound in vanilla, is primarily derived from mature pods of vanilla (*Vanilla planifolia* Andrews). Although the biosynthetic pathway of vanillin has been progressively elucidated, the specific key enzymes and transcription factors (TFs) governing vanillin biosynthesis require further comprehensive investigation via combining transcriptomic and metabolomic analysis. For this study, *V. planifolia* (higher vanillin producer) and *V. imperialis* (lower vanillin producer) were selected. Time-series metabolomics analysis revealed 160–220 days after pollination (DAPs) as the critical phase for vanillin biosynthesis. Combined time-series transcriptome analysis revealed 984 upregulated differentially expressed genes (DEGs) in key periods, 2058 genes with temporal expression, and 4326 module genes through weighted gene co-expression network analysis (WGCNA), revealing six major classes of TFs: No Apical Meristem (NAC), Myb, WRKY, FLOWERING PROMOTING FACTOR 1-like (FPFL), DOF, and PLATZ. These TFs display strong regulatory relationships with the expression of key enzymatic genes, including P450s, COMT, and 4CL. The NAC TF family emerged as central regulators in this network, with *NAC-2* (*HPP92_014056*) and *NAC-3* (*HPP92_012558*) identified as key hub genes within the vanillin biosynthetic gene co-expression network. The findings of this study provide a theoretical foundation and potential target genes for enhancing vanillin production through genetic and metabolic engineering approaches, offering new opportunities for sustainable development in the vanilla industry and related applications.

## 1. Introduction

Vanilla is a tropical orchid, known as the “king of food and spices” [1,2], which is the second most expensive spice in the world and is one of the most important and popular aromatic compounds used in food, beverages, and cosmetics [3,4,5,6]. Vanillin, as the most valuable chemical component of vanilla, is of undeniable importance in the food and flavoring industries [7]. Vanillin not only imparts the distinctive aroma to vanilla but is also widely used as a source of natural food additives and flavorings. However, the complexity of vanillin’s biosynthetic pathway and the high cost of extraction have driven researchers to seek more efficient synthetic methods. *V. planifolia* is the principal source of natural vanillin, with *V. planifolia* Andrews and *V. imperialis* Kraenzl being two widely cultivated species in China [8]. Research has found that the vanillin content in *V. planifolia* is significantly higher than that in *V. imperialis* [9,10].

The synthetic pathway of vanillin involves a complex biotransformation process, primarily involving the conversion of coumarin and vanillin aldehyde (Appendix A) [11,12,13,14,15,16,17,18,19,20], a process meticulously regulated by various enzymes, including vanillin oxidase and vanillic acid hydroxylase [21,22]. These enzymes and intermediate metabolites (e.g., glucovanillin) are strictly regulated by genetic factors and are crucial for the yield and quality of vanillin [16,23]. Currently, research on improving vanillin synthesis and production is moving towards a diversified and integrated approach [14,24,25]. Researchers are actively combining molecular biology, metabolic engineering, synthetic biology, and traditional plant breeding techniques to enhance the yield and quality of vanillin [26,27,28]. For instance, gene editing techniques, such as targeting the vanillin synthase gene, can precisely regulate the expression of key enzymes involved in vanillin synthesis, thereby increasing its synthetic efficiency [29,30,31]. Simultaneously, the potential of vanillin synthesis in other plants and microorganisms is also being explored. This includes the use of genetically modified plants, microbial fermentation, and synthetic biology methods to optimize metabolic pathways and identify new sources of vanillin production [16,32].

The biosynthetic pathway of vanillin is complex and involves multiple steps, primarily including the conversion of *p*-coumaric acid and vanillic acid to vanillin [16,22]. *p*-coumaric acid is converted to vanillic acid by the action of vanillin synthase, after which vanillic acid is further transformed into vanillin [33,34,35]. Additionally, some studies indicate that ferulic acid is generated through a conversion pathway involving cinnamic acid and other intermediates, ultimately leading to the formation of vanillin [36]. Feruloyl esterases can hydrolyze ferulic acid esters, releasing ferulic acid and indirectly influencing vanillin synthesis [36,37,38]. Transcription factors play significant regulatory roles in the vanillin biosynthesis process. For instance, the *MuYqhC* transcription factor variant assists in screening for the efficient COMT enzyme variant *Mu176* by constructing a biosensor system, significantly enhancing the synthesis efficiency of vanillin from ferulic acid [39]. Current research seldom mentions transcription factors involved in the plant biosynthesis of vanillin, which may be attributed to the more complex nature of vanillin synthesis pathways and regulatory mechanisms in plants compared to microorganisms, involving multilayered biochemical and genetic controls [35,40,41].

This study aims to address this research gap by systematically investigating the transcription factors and the regulatory mechanisms involved in vanillin synthesis. *V. planifolia* and *V. imperialis*, which originated in the rainforests of Central and South America, have been successfully domesticated in China, and it has been found that *V. planifolia* contains significantly higher vanillin content than *V. imperialis*. Through integrated metabolic profiling and time-series transcriptome sequencing of pod samples from both plant species, we conducted systematic investigations focused on the vanillin biosynthesis pathway.

## 2. Results

### 2.1. Overview of Vanillin Accumulation Patterns During Vanilla Species Development

Metabolomic data of high-vanillin *V. planifolia* across six developmental time points (30, 40, 60, 100, 160, and 220 DAPs, designated as X) and low-vanillin *V. imperialis* (30 and 40 DAPs, designated as V) were used. The experimental approach incorporated rigorous quality-controlled UPLC-MS/MS analyses, where total ion chromatograms demonstrated stable baselines with excellent peak intensity and retention time consistency, confirming optimal instrument performance (Appendix A). A principal component analysis (PCA) of the quality control samples revealed excellent reproducibility (Appendix A). The correlation between the QC samples, with correlation coefficients ≥ 0.99 among five biological replicates, significantly exceeded technical requirements (Appendix A). These results provide evidence for the reliability of the metabolite detection.

Glucovanillin serves as the inactive glycosylated precursor of vanillin, playing an essential role as a key intermediate metabolite in vanillin biosynthesis. Glucovanillin is a glycosylated and biologically inactive compound requiring enzymatic hydrolysis to release active vanillin, which possesses flavor/aroma properties [7,10]. The analysis of the glucovanillin content dynamics and corresponding fold-change patterns revealed three distinct phases of glucovanillin accumulation in developing vanilla pods: The initial accumulation phase (30–40 DAPs) showed minimal increase, with X40 displaying only a 1.53-Log2FC enhancement compared to V40, indicating a gradual activation of early biosynthetic processes. During the stabilization accumulation phase (60–100 DAPs), the glucovanillin content showed a significant and progressive increase relative to the V30 baseline. The accumulation began with a marked 13.08-Log2FC increase (X60) compared to V30, indicating a pivotal metabolic shift toward active vanillin biosynthesis. This upward trend continued systematically through subsequent sampling intervals: X100 (18.96-Log2FC) (Figure 1A, Appendix A). The sustained rise in the glucovanillin content underscores a robust and progressively enhanced biosynthetic capacity during this developmental window. The metabolic rapid accumulation phase (160–220 DAPs) revealed peak accumulation with X160 (20.85-Log2FC) and X220 (21.76-Log2FC), where the glucovanillin content exhibited dramatic elevation, indicating the establishment of a metabolic steady state while maintaining maximum biosynthetic activity across all of the analyzed time points in our study. Furthermore, based on the dynamic changes in vanillin content, we established the 160–220 day after pollination (DAP) period as the most critical stage for vanillin biosynthesis in *Vanilla planifolia* (Appendix A).

Further, transcriptional analysis identified the vanillin synthase gene *VpVAN* (*HPP92_026221*) as a key candidate in the vanillin biosynthetic pathway, exhibiting an expression profile that tightly mirrored metabolic dynamics (Figure 1B). Notably, *VpVAN* maintained consistently high transcript levels during the 160–220 DAP period, precisely coinciding with the phase of rapid vanillin accumulation. These results provide valuable temporal markers for vanilla quality improvement, especially the 160–220 DAP period, suggesting that future molecular breeding strategies should focus on optimizing the expression of key biosynthetic enzymes during these critical developmental windows to enhance vanillin production potential. This developmental-stage-specific approach may serve as a model for improving secondary metabolite production in other aromatic plant species.

### 2.2. DEG Analysis Provides Molecular Insights into Vanillin Biosynthesis

We performed transcriptome sequencing analysis on nine samples comprising *V. imperialis* (30 and 40 DAPs) and *V. planifolia* (30, 160, and 220 DAPs) using high-throughput sequencing technology. A total of 709.33 Gb of high-quality sequencing data was obtained, with an average sequencing depth exceeding 26 Gb per sample. Quality assessment validated the reliability of the experimental results through both principal component analysis (PCA) and inter-sample Pearson correlation analysis. In PCA dimensionality reduction, *V. imperialis* samples (V30 and V40), characterized by lower vanillin content, distinctly clustered away from *V. planifolia* samples along the first and second principal components, while the two *V. imperialis* developmental stages (V30 and V40) maintained tight clustering (Appendix A). This separation reflects species-specific expression patterns of genes involved in vanillin biosynthesis. Notably, Pearson correlation analysis revealed a strong transcriptional similarity (r > 0.9) between *V. imperialis* samples at different developmental stages (V30 vs. V40), further confirming the stability of key vanillin biosynthetic gene expression in this species (Appendix A). The dataset of 709.33 Gb of transcriptomic data meets international quality control standards for transcriptome research, providing a robust foundation for subsequent molecular analyses.

This study systematically investigated the transcriptional regulation features during the rapid vanillin accumulation phase through comparative transcriptome analysis of *V. imperialis* pods at different developmental stages. Using 30 DAP (V30) and 40 DAP (V40) pods with low vanillin content and 30 DAP control pods (X30) as references, we performed six pairwise differential expression analyses against pods at 160 DAPs and 220 DAPs (rapid accumulation phase): V30_X160, V30_X220, V40_X160, V40_X220, X30_X160, and X30_X220 (Figure 1A). Differential expression analysis identified 11,658, 11,392, 11,481, 11,392, 6764, and 6518 differentially expressed genes (DEGs) in each comparison, respectively; the total number of genes is 29,168. Intersection analysis revealed 2576 common DEGs across all comparisons. Gene Ontology (GO) enrichment analysis demonstrated significant enrichment of these common genes in fundamental biological processes including cell wall biogenesis (GO:0009832), external encapsulating structure formation (GO:0030312), carbohydrate metabolic processes (GO:0005975), auxin response (GO:0009733), glycosyltransferase activity (GO:0016757), root morphogenesis (GO:0010015), and cellular carbohydrate metabolism (GO:0044262), all of which are closely associated with plant growth and development (Appendix A). Concurrently, we identified 4817, 4767, 4609, 4657, 4024, and 3916 downregulated genes in the respective comparisons, with 1247 genes common across all groups (Figure 2A,B). These downregulated genes were primarily enriched in pathways related to cell wall formation, the external encapsulating structure, and the cytoskeleton (GO:0005856), suggesting a progressive decline in morphogenesis-related gene expression during late pod development stages (Appendix A).

Significantly, the analysis identified 6841, 6625, 6872, 6735, 2740, and 2602 upregulated genes in each comparison, respectively, with 984 genes consistently upregulated across all groups (Figure 2C). GO analysis indicated that these upregulated genes were predominantly enriched in secondary metabolite biosynthetic processes (GO:0044550) and phenylpropanoid metabolic process (GO:0009698). Importantly, this set contained several key enzymatic genes involved in the phenylpropanoid pathway, including flavanone 3-hydroxylase (F3H), cinnamoyl-CoA reductase 2 (CCR2), cinnamyl alcohol dehydrogenase (CAD), and 4-coumarate-CoA ligase (4CL), all of which play pivotal catalytic roles in secondary metabolite biosynthesis (Figure 2D, Appendix A). These findings collectively demonstrate that during pod development, there is a gradual decrease in morphogenesis-related gene expression concomitant with progressive enhancement of genes associated with vanillin biosynthesis. To validate the RNA-seq data, quantitative PCR (qPCR) analyses were performed on some DEGs. The qPCR results confirmed the reliability of the identified DEGs (Appendix A).

### 2.3. Temporal Expression Patterns of Vanillin-Related Genes in Vanilla Species

This study conducted a systematic analysis of 1605 metabolites in *V. planifolia* pods across six developmental stages (30, 40, 60, 100, 160, and 220 DAPs), categorizing them into 10 distinct metabolite clusters using a fuzzy C-means clustering algorithm. Based on previous studies, it has been established that glucovanillin, a metabolite, is the primary precursor in the synthesis of vanillin; we found it belonged to Cluster 2, showing a characteristic triphasic accumulation pattern: stable levels during early development (30–40 DAPs), a rapid increase from 60–100 DAPs, and peak accumulation at 160–220 DAPs (Figure 3A). This indicates that the genes in Cluster 9 may play a role in regulating the synthesis of vanillin (Figure 3B).

Parallel transcriptomic analysis of the same developmental stages identified 10 gene clusters (Clusters 1–10) containing 3270, 2449, 2306, 2970, 2265, 3316, 2257, 3374, 1945, and 1569 genes, respectively, through fuzzy C-means clustering. GO analysis revealed distinct functional specialization between clusters: Cluster 1 genes were enriched in nucleolar function (GO:0005730) and ribonucleoprotein complex biogenesis (GO:0022613), Cluster 2 in vesicle-mediated transport (GO:0016192) and cellular macromolecule localization (GO:0070727), Cluster 3 in ribonucleoprotein complex biogenesis (GO:0022613) and ncRNA metabolic process (GO:0034660), Cluster 4 in protein modification by small protein conjugation (GO:0070647) and the protein catabolic process (GO:0030163), Cluster 5 in intracellular signal transduction (GO:0035556) in the cell wall (GO:0005618), Cluster 6 in anion binding (GO:0043168) and enzyme binding (GO:0019899), Cluster 7 in plasma membrane components (GO:0044459) in the chromosome (GO:0005694), Cluster 8 in organophosphate metabolism (GO:0019637) and cellular response to DNA damage stimulus (GO:0006974), Cluster 9 in secondary metabolic processes (GO:0019748) and plant organ morphogenesis (GO:1905392), and Cluster 10 in cellular response to acid chemicals (GO:0071229). Notably, among the ten identified clusters, Cluster 9 stood out as the only one showing the most significant enrichment of genes involved in secondary metabolite biosynthesis. This cluster was uniquely characterized by containing the key vanillin biosynthesis regulatory gene HPP92_026221, demonstrating an exceptional temporal expression pattern that exhibited remarkable synchrony with glucovanillin accumulation dynamics in metabolite Cluster 2. Both Cluster 9 gene expression and Cluster 2 glucovanillin accumulation displayed identical developmental trajectories—maintaining stable levels during early development (30–40 DAPs), undergoing a rapid increase from 60–100 days, and eventually entering a period of high expression between 160 and 220 days, showcasing a highly coordinated pattern that suggests Cluster 9 likely plays a central regulatory role in vanillin biosynthesis pathway activation.

Detailed GO enrichment analysis of Cluster 9 revealed key pathways predominantly involved in carbon metabolism, amino acid biosynthesis, and phenylpropanoid biosynthesis (Appendix A). These pathways are crucial for the biosynthesis and regulatory mechanisms of vanillin, a principal compound originating from the phenylpropanoid pathway, underscoring its vital role. Concurrently, the GO analysis identified enrichment in biological processes, including secondary metabolic processes, responding to antibiotics, which may involve transcriptional regulators known to modulate phenylpropanoid metabolism, and monocarboxylic acid metabolic processes, which involve vanillic acid, the immediate precursor for vanillin. These processes are indicative of the complex biological framework supporting vanillin biosynthesis and its multifaceted roles within the organism. Cluster 9 emerged as the priority for subsequent analysis, as this gene group directly regulates the accumulation of key metabolites in the vanillin biosynthesis pathway.

### 2.4. Co-Expression Networks and Hub Genes of Vanillin Biosynthesis

To elucidate the molecular regulatory mechanisms governing vanillin biosynthesis, this study employed an integrated computational approach using weighted gene co-expression network analysis (WGCNA) based on pre-existing transcriptomic data. Differential gene expression analysis was performed with a stringent threshold of fold change > 2 to identify meaningful transcriptional variations. During computational parameter optimization, we systematically evaluated soft threshold values ranging from 1 to 20 in relation to scale-free topology fitting indices (Appendix A). The optimal soft threshold was determined to be β = 10 when the scale-free topology fit index reached 0.8 (Appendix A), ensuring the construction of a biologically relevant co-expression network with distinctive scale-free properties.

Network topology evaluation confirmed robust intramodular connectivity patterns across major network components, validating the computational reliability of our analytical framework. Utilizing a dynamic tree-cutting algorithm, we efficiently partitioned genes into functionally cohesive modules, with the hierarchical clustering dendrogram clearly delineating intergenic relationships (Figure 4A). Following module consolidation at a 90% similarity threshold, seven distinct co-expression modules were identified that exhibited significant biological characteristics (Figure 4B). The gray module contained the highest number of differentially expressed genes (7152), whereas the purple module comprised the fewest (713). Notably, each module displayed specific gene composition patterns, suggesting that distinct functional gene clusters may be coordinately regulated through modular mechanisms during the developmental progression of vanilla pods.

Subsequently, metabolites involved in vanillin synthesis were used as trait indicators, and a correlation analysis was conducted with the seven modules. The results, depicted in Figure 4B, show a significant positive correlation between the green module and several metabolites, including vanillyl alcohol (R = 0.86, *p* < 0.001), glucovanillin (R = 0.89, *p* < 0.001), vanillin (R = 0.95, *p* < 0.001), vanillic acid (R = 0.92, *p* < 0.001), chlorogenic acid (R = 0.86, *p* < 0.001), *p*-coumaric acid (R = 0.84, *p* < 0.001), cinnamic acid (R = 0.82, *p* < 0.001), 3,4-Dihydroxybenzoic acid (R = 0.90, *p* < 0.001), and coniferyl alcohol (R = 0.94, *p* < 0.001). Notably, the correlation between the green module and vanillin was the highest. The gene–module membership correlation coefficient reached 0.92 in the green module, indicating a strong association between the genes within this module and vanillin synthesis (Figure 4C). In GO terms, the genes within MEgreen primarily respond to metal ions (GO:0010038), which may involve structural stabilization of carboxylic acid reductase (CAR), reducing vanillic acid and other vanillin precursors [34] or other mechanisms [28]; cation binding (GO:0043169), which may mediate the transmembrane transport of vanillin precursor [42]; and monocarboxylic acid metabolic process (GO:0032787), which may mediate the cellular uptake of lactate and other metabolic precursors [43] (Appendix A), which establishes metabolic connections with vanillin biosynthesis through precursor supply and coenzyme coordination mechanisms.

Building upon the established co-expression network associated with vanillin biosynthesis, we conducted systematic hub gene identification within the green module using the Maximum Clique Centrality algorithm. The analysis revealed a set of ten pivotal genes, comprising nine functional protein-coding genes and one transcription factor worth special attention (Figure 4D). Among these, *HPP92_012558* emerged as a NAC family transcription factor exhibiting central regulatory importance within the network architecture. Functional annotation of these hub genes demonstrated their involvement in several critical metabolic pathways: the transketolase-encoding *HPP92_025679* participates in the pentose phosphate pathway; *HPP92_024211* encodes adenosylhomocysteinase, influencing methylation metabolism; *HPP92_014452* represents an α-trehalose glucohydrolase involved in carbohydrate metabolic regulation; *HPP92_014557* encodes a peroxidase-related protein mediating oxidative stress response; and *HPP92_014846* corresponds to 6-phosphogluconate dehydrogenase, a key enzyme in NADPH generation. These metabolic regulators were complemented by other functionally significant elements including *HPP92_004378*, which encodes phospholipid:diacylglycerol acyltransferase 1 (PDAT1) in phospholipid metabolism, along with *HPP92_013950* and *HPP92_009971*, which are involved in amino acid metabolic pathways.

The identified hub genes demonstrate extensive interconnectivity within the metabolic network, suggesting their potential to directly or indirectly modulate vanillin biosynthesis through either secondary metabolic flux regulation or transcription factor network interactions. Notably, the discovery of these strategic regulators in the green module provides valuable molecular targets for subsequent investigations aiming to unravel the intricate regulatory mechanisms controlling vanillin production efficiency at the biosynthetic level.

### 2.5. Key Genes Responding to Regulatory Network in Vanillin Biosynthesis

To systematically elucidate the molecular regulatory network underlying vanillin biosynthesis, we performed an integrative analysis combining data from 984 upregulated differentially expressed genes (DEGs), 1945 genes from temporal expression Cluster 9, and 4325 genes from the WGCNA green module. This multidimensional approach identified 433 candidate genes, among which we annotated 18 functional proteins directly involved in the vanillin biosynthetic pathway (Figure 5A). These proteins were categorized into seven functional classes, collectively constituting the complete metabolic network from phenylpropanoid biosynthesis to vanillin production.

The pathway analysis revealed critical regulatory nodes, beginning with the rate-limiting enzyme catalyzing the conversion of *p*-coumaric acid to caffeoyl-CoA in the initial phenylpropanoid metabolism phase. Subsequent methylation reactions were mediated by caffeoyl-CoA O-methyltransferase, generating feruloyl-CoA, a metabolic branch point directing flux toward either lignin or vanillin precursors. Cytochrome P450 family members (including C4H and F5H) were identified as key players in aromatic ring hydroxylation, influencing intermediate product distribution. In the vanillin-specific branch, caffeic acid O-methyltransferase (COMT) emerged as the central enzyme catalyzing ferulic acid conversion to vanillin. Members of the short-chain dehydrogenase/reductase (SDR) family were implicated in aldehyde group redox regulation, potentially controlling the vanillin-to-vanillic acid conversion efficiency. Notably, flavin-containing monooxygenases (FMOs) and UDP-glycosyltransferases (UGTs) were identified as regulators of ring hydroxylation and vanillin glycosylation, respectively, with UGT activity being particularly crucial for stabilizing vanillin derivatives.

The characterization of these 18 functional proteins not only delineates the complete enzymatic toolkit for vanillin biosynthesis but, more importantly, identifies critical regulatory nodes at metabolic branch points. The coordinated action of COMT and UGT appears particularly significant for controlling both vanillin yield and storage stability. These findings provide a solid theoretical foundation and target selection strategy for future metabolic engineering approaches (Appendix A).

Further, 23 transcription factors (TFs) potentially involved in vanillin biosynthesis regulation were identified, focusing particularly on the 10 most differentially expressed members from the NAC (4 members), DOG, FPFL (FLOWERING PROMOTING FACTOR 1-like), HSF, Myb, PLATZ, and WRKY families. To elucidate their regulatory relationships with vanillin biosynthesis structural genes, we constructed an interaction network through correlation analysis visualized using Cytoscape (V3.9.1) (Figure 5B).

Network topology analysis based on Maximum Clique Centrality (MCC) and closeness algorithms revealed that *NAC-2* (*HPP92_014056*) emerged as a central hub gene within the network. *NAC-2* exhibited strong associations with multiple key vanillin biosynthetic genes, including two *4CLs* (4-coumarate-CoA ligases), two *CCoAOMTs* (caffeoyl-CoA O-methyltransferases), one *COMT* (caffeic acid O-methyltransferase), one *FMO* (flavin-containing monooxygenase), nine cytochrome P450 genes (*CYP450s*), one *SDR* (short-chain dehydrogenase/reductase), and one *UGT* (UDP-glycosyltransferase). Strikingly, *NAC-2* and *COMT* showed a high Pearson correlation coefficient (r = 0.944), suggesting their tight functional coupling in vanillin regulation. Additionally, *NAC-2* correlated with nine hydroxylase-encoding P450 genes (r = 0.716–0.889), indicating its potential role in modulating hydroxylation and oxidative modification steps in phenylpropanoid metabolism (Appendix A).

Furthermore, another NAC family member, *NAC-3* (*HPP92_012558*), exhibits a remarkably strong co-expression relationship (r = 0.924) with COMT. Significantly, *NAC-3* serves not only as a crucial transcriptional regulator but also emerges as a key hub gene within the green module through WGCNA, representing the sole transcription factor identified in this module. These findings substantially reinforce the pivotal role of NAC family members in the regulatory network underlying vanillin biosynthesis, particularly *NAC-2* (*HPP92_014056*) and *NAC-3* (*HPP92_012558*).

## 3. Discussion

This study systematically identified the key regulatory network underlying vanillin biosynthesis during pod development in tropical vanilla (*Vanilla* spp.) cultivated in China through integrated metabolomic and transcriptomic analyses. Our findings demonstrate that NAC family transcription factors, particularly *NAC-2* (*HPP92_014056*) and *NAC-3* (*HPP92_012558*), mediate metabolic reprogramming during peak vanillin production by regulating the expression of the P450 and COMT gene families.

Molecular evidence suggests that NAC transcription factors likely directly modulate the expression levels of phenylpropanoid pathway genes (including PAL, C4H, and 4CL) through binding to cis-regulatory elements in their promoter regions [44,45]. This regulatory mechanism exhibits that NACs may potentially regulate many genes involving vanillin synthase to enhance precursor conversion. Furthermore, protein–protein interactions between NACs and other transcription factors (such as MYB and bHLH) may form regulatory complexes that amplify metabolic control [44,46]. Conservation of this regulatory paradigm is evident across plant species. In Eucommia ulmoides, the *EuNAC* subgroup influences ferulic acid (a vanillin precursor) availability by modulating lignin biosynthesis genes like *LACCASES* [47]. NAC-MYB synergistic regulation of lignin pathway genes has been documented in lianas [48,49]. Ginkgo biloba demonstrates indirect NAC-mediated stimulation of vanillin synthesis through flavonoid pathway crosstalk with phenylpropanoid metabolism [50]. *V. planifolia* high-vanillin cultivars exhibit differential expression of phenylpropanoid genes with *NACs* serving as central regulatory nodes [15]. Phytohormonal signals (e.g., jasmonate and ethylene) may indirectly modulate vanillin production via *NAC* and *AP2/ERF* transcriptional regulation [51,52,53,54,55].

The cytochrome P450 enzyme family exhibits a close functional relationship with vanillin biosynthesis in plants. As crucial players in various secondary metabolic pathways, P450 enzymes actively participate in phenylpropanoid and flavonoid biosynthesis—both of which generate key vanillin precursors such as coniferyl alcohol and ferulic acid [52,53]. In *V. planifolia* (the primary vanilla-producing species), transcriptomic analyses have identified differentially expressed P450 genes (e.g., CYP81A subfamily members) that show strong correlation with vanillin accumulation patterns. These P450 genes likely influence vanillin production by modulating flux through the phenylpropanoid pathway, which provides essential precursors for vanillin biosynthesis. While the final vanillin biosynthesis step (conversion of ferulic acid to vanillin) has traditionally been attributed to *VpVAN* (vanillin synthase), recent biochemical studies reveal that certain P450 monooxygenases (including engineered microbial dioxygenases) can catalyze this transformation in a single enzymatic step [56,57]. This finding expands our understanding of alternative enzymatic routes for vanillin production in both plant and microbial systems.

Research has identified MYB transcription factors as crucial regulators in vanillin biosynthesis in Vanilla planifolia. These factors potentially optimize vanillin production efficiency through direct or indirect modulation of the phenylpropanoid pathway and by forming transcriptional complexes with other regulators (e.g., WRKY and NAC factors). Specifically, R2R3-MYB transcription factors influence vanillin precursor synthesis by precisely controlling the phenylpropanoid pathway [58,59]. A particularly significant finding reveals that MYB factors frequently interact with bHLH and WD40 proteins to form MBW (MYB-bHLH-WD40) complexes [60,61], which are known to play pivotal regulatory roles in anthocyanin biosynthesis. Given the shared precursor substrates between vanillin and anthocyanin biosynthesis within the phenylpropanoid pathway, we propose that MYB may employ an analogous complex mechanism to activate pathway-specific genes involved in vanillin production [62]. Comparative analysis of materials with varying vanillin contents demonstrated significantly elevated MYB expression levels in high-vanillin samples, strongly suggesting its role as a positive regulator of vanillin biosynthesis.

The WRKY transcription factors serve as pivotal regulators in vanillin biosynthesis, primarily influencing vanillin accumulation through indirect modulation of key precursors in the phenylpropanoid pathway [63,64,65]. These factors potentially exert precise control over vanillin metabolism by integrating defense responses, hormonal signals, and protein–protein interactions with other regulators such as MYB and OBE. WRKY interaction with OBE proteins may optimize phenylpropanoid metabolism while maintaining normal growth processes, potentially through stress-related gene regulation [66,67]. A parallel regulatory mechanism is observed in *Vanilla planifolia*, where *VpVAN* overexpression substantially enhances vanillin production, a process that may involve coordinated regulation by WRKY in conjunction with other transcription factors like MYB [29].

SDR, a NAD(H)-dependent oxidoreductase superfamily, may indirectly influence vanillin biosynthesis by modulating terpenoid metabolism, which shares carbon sources and functional genes with the phenylpropanoid pathway, which is the primary route for vanillin precursor synthesis [68]. The role of SDR in NADPH-dependent oxidoreduction implies it might influence precursor (e.g., ferulic acid, coniferyl alcohol) availability by altering redox balance or upstream pathway flux [68]. Notably, SDR-mediated modifications of terpenoids could indirectly divert metabolic resources toward phenylpropanoid-derived vanillin synthesis, as these pathways likely crosstalk at the level of shared intermediates or cofactor pools [15]. Moreover, the CYP450 family participates in terpenoid modification (such as oxidation) [69], while vanillin biosynthesis also relies on CYP450 enzymes, for example, *p*-coumaric acid hydroxylation [21]. These two processes may be indirectly connected through shared CYP450 isoforms or regulatory networks.

Our findings reveal that 4CL, P450, COMT, and SDR play indispensable roles in vanillin biosynthesis, providing deeper molecular insights beyond previous reports [15]. Further, WGCNA unveiled the transcriptional network regulating vanillin production, identifying core modules where NAC, MYB, and WRKY transcription factors showed the strongest correlations with biosynthetic genes. Notably, NAC members recurrently appeared across modules, suggesting their pivotal regulatory role in vanillin synthesis.

Our research provides novel insights into the molecular governance of vanillin biosynthesis, particularly elucidating the pivotal role of NAC transcription factors in tropical vanilla cultivars. The identified *NAC-2* (*HPP92_014056*) and *NAC-3* (*HPP92_012558*) family members present precise molecular targets for breeding programs. Subsequent development of associated molecular markers could expedite varietal improvement for enhanced vanillin content, thereby elevating the economic competitiveness of China’s spice crops. Moreover, the characterized NAC regulatory network offers a framework for investigating secondary metabolite biosynthesis in other cash crops.

## 4. Materials and Methods

### 4.1. Plant Materials and Treatment

Vanilla pods were collected at 30, 40, 60, 100,160, and 220 days after pollination (DAPs) from Spice and Beverage Research Institute, Chinese Academy of Tropical Agricultural Sciences, Xinglong, Hainan Province, China (110°20′ E, 18°73′ N). The region is characterized by a mean annual temperature of 24.5 °C and an average annual precipitation of 2201 mm. The *Vanilla plantations* have been operational for 6 years. The soil contained 15.1 g kg^−1^ of organic matter, 0.80 g kg^−1^ of total N, 21.9 mg kg^−1^ of available P (Olsen), 92.3 mg kg^−1^ of available K (NH4OAc), and a pH of 6.56 (1:1, soil:water ratio). The annual fertilizer application was 120 kg N ha^−1^ (as urea), 50 kg P ha^−1^ (as superphosphate), and 90 kg K ha^−1^ (as muriate of potash). Vanilla naturally resists pests, requiring no chemical pesticide spraying. Vanilla orchids were cultivated on 1.5-m-high trellises at 1.2 m × 1.8 m spacing. Vanilla initiates flowering in April with subsequent hand-pollination. Pod samples of *V. imperialis* at 30 and 40 DAPs were selected for both metabolic profiling and transcriptomic analysis; *V. planifolia* pods were harvested at 30, 40, 60, 100, 160, and 220 DAPs for metabolic profiling and transcriptomic analysis. This experiment was carried out from May to December in 2023. Pod samples of *V. imperialis* at 30 and 40 DAPs were selected for both metabolic profiling and transcriptomic analysis; *V. planifolia* pods were harvested at 30, 40, 60, 100, 160, and 220 DAPs for metabolic profiling and transcriptomic analysis.

For both varieties, 3 pods from each time-point sample were randomly collected and immediately transported to the laboratory on ice. After being sliced open with sterile knives, a portion of the samples was immediately frozen in liquid nitrogen and stored at −80 °C for subsequent RNA extraction. Methanol (MeOH), acetonitrile (ACN), and ethanol (chromatographic purity grade) used for LC-MS analysis were purchased from Merck. Other extraction and analytical reagents (analytical grade) were obtained from Shanghai Chemical Reagents Co., Ltd. (Shanghai, China).

### 4.2. Metabolite Analysis

#### 4.2.1. Metabolite Extraction

A tissue sample of 100 mg lapped with liquid nitrogen was placed in an EP tube containing 500 mL of an 80% methanol aqueous solution. The sample was vortex shaken, bathed in ice for 5 min, and centrifuged at 15,000× *g* and 4 °C for 20 min; a certain amount of supernatant was taken, diluted with water by mass spectrometry to the methanol content of 53%, and centrifuged at 15,000× *g* and 4 °C for 20 min. The supernatant was then collected and injected into LC-MS for analysis.

#### 4.2.2. Metabolite Detection and Chromatographic Condition Analysis

A HypesilGold column (C18, 100 × 2.1 mm, 1.9 mm) was used to sample 2 µL at a flow rate of 0.2 mL/min and a column temperature of 40 °C with an automatic injector set to 8 °C. The positive and negative modes were adopted, with 0.1% formic acid as the positive moving phase A and methanol as the moving phase B. The negative mode moving phase A is 5 mM ammonium acetate, pH 9.0, and the moving phase B is methanol. The gradient elution procedure is as follows: 0–1.5 min, 98–15% A, 2–85% B; 1.5–3 min, 0–15% A, 85–100% B; 3–10 min, 0–98% A, 100–2% B; 10–10.1 min, 98% A, 2% B; and 11–12 min, 98% A, 2% B. The mass spectrum conditions are as follows. The scanning range is 100–1500 *m*/*z*. The ESI source setup includes a 3.5 kV spray voltage, 35 psi sheath gas flow rate, auxiliary gas flow rate of 10 L/min, 320 °C ion transfer tube temperature (capillary temperature), 60 ion input RF level (S-lens RF level), 350 °C auxiliary gas heater temperature, and positive/negative polarity. The MS/MS secondary scan is a data-dependent scan.

#### 4.2.3. Metabolite Data Preprocessing and Metabolite Identification

The original files were imported into the CD 3.1 library retrieval software for processing, and the retention time, mass-charge ratio, and other parameters of each metabolite were screened. A retention time deviation of 0.2 min and mass deviation of 5 ppm were set for peak alignment of the different samples, and then peak extraction and peak area quantification were performed. Then, the target ions were integrated, and the molecular formula was predicted using molecular ion peaks and fragment ions and compared with the mzCloud (https://www.mzcloud.org/), mzVault, and Masslist databases. After standardized treatment, the relative peak area was obtained.

Compounds with CVs greater than 30% of the relative peak area were removed from QC samples. Finally, the identification and relative quantitative results of metabolites were obtained.

### 4.3. Transcriptome Analysis

Pod samples were collected from *V. planifolia* at 30, 40, 60, 100, 160, and 220 DAPs and from *V. imperialis* at 30 and 40 DAPs, with three biological replicates for each developmental stage. They were subjected to RNA isolation and purification using the RNAprep Pure Plant Plus kit (TIANGEN BIOTECH (BEIJING) Co., Ltd., Beijing, China). Total RNA quality was assessed using the Fragment Analyzer 5400 (Agilent Technologies, Santa Clara, CA, USA). cDNA libraries were constructed using NEBNext^®^ UltraTM RNA Library Prep Kit for Illumina^®^ (New England Biolabs, Ipswich, MA, USA) and subjected to sequencing. The library formulation was sequenced using the Illumina Novaseq 6000 platform (Novogene Bioinformatics Technology Co., Ltd., Beijing, China) to generate paired-end reads of 150 bp. Raw data converted from the Illumina platform was subjected to quality control before subsequent analysis.

Raw sequencing data were first subjected to quality control using fastp (v0.23.2) [70] with default parameters to obtain high-quality reads. Subsequently, the filtered reads were aligned to the *V. planifolia* reference genome (PRJNA633886) using HISAT2 (v2.1.0) [71] with the parameters ”-p 8 -dta”. Gene and transcript expression levels were then quantified as FPKMs (Fragments Per Kilobase Million) values through featureCounts (v2.0.3) [72], employing the parameters ”-p -t CDS -g gene_id”. Genes with FPKMs > 1 were retained for further analysis, and the selected expression data were log2(FPKM + 1)-transformed before clustering analysis and visualization via the R pheatmap package (V4.1.3). Additionally, DESeq2 (v1.46.0) was used to calculate differential expression across three biological replicates. Differentially expressed genes (DEGs) were identified based on the threshold of |log2FC| > 1 with a false discovery rate (FDR) < 0.05. Temporal expression pattern analysis was performed using Mfuzz to identify genes showing stage-specific expression dynamics during pod development [73].

### 4.4. RNA Isolation and Quantitative RT-PCR

To investigate the expression patterns of differentially expressed genes (DEGs), the vanilla pods were sampled at six developmental stages (30, 40, 60, 100, 160, and 220 days after pollination, DAPs) for quantitative reverse transcription PCR (qRT-PCR) analysis, with three biological replicates collected per time point. Total RNA was isolated using the TIANGEN RNA Plant Kit, followed by cDNA synthesis with the Vazyme Reverse Transcription Kit. qRT-PCR amplification was conducted on an ABI 7500 Real-Time PCR System using Vazyme’s SYBR Green Master Mix, with ACTIN serving as the internal control gene. Relative gene expression levels were calculated using the 2^−ΔΔCT^ method. Primers are provided in Appendix A.

### 4.5. Combined Transcriptome and Metabolome Analysis

To elucidate the genetic regulation underlying vanillin biosynthesis, we performed weighted gene co-expression network analysis (WGCNA), integrating both metabolic substrates and transcriptomic data [74]. Complementary metabolomic profiling was conducted to identify vanillin-associated metabolites that co-expressed with the MEgreen module identified through the WGCNA. Correlation analysis was performed on 16 metabolites related to vanillin content and the obtained modules, with a correlation coefficient > 0.75 used as the screening criterion for candidate modules. Hub genes were selected based on gene significance (GS) > 0.5 and module membership (MM) > 0.8 (with MM > 0.9 in the green module) as criteria. Using cytoHubba algorithms, we identified ten hub genes within the MEgreen module, which were subsequently subjected to comprehensive functional annotation through the pfam, COG, KOG, GO, and KEGG databases. This systematic approach revealed six distinct classes of transcription factors potentially involved in regulatory networks. To investigate functional relationships between these transcription factors and vanillin biosynthesis pathway metabolites, we calculated pairwise Pearson correlation coefficients and constructed regulatory networks visualized using Cytoscape [75].

## 5. Conclusions

This research focused on two successfully cultivated vanilla varieties in China’s tropical regions, employing integrated metabolomic and transcriptomic approaches to investigate key biochemical changes during fruit capsule development and elucidate the molecular mechanisms governing vanillin biosynthesis.

Metabolomic profiling and vanillin synthase gene expression patterns revealed that vanillin synthesis reached its peak between 160 and 220 DAPs. Transcriptomic comparative analysis across six developmental time points identified 984 consistently upregulated genes, with GO enrichment analysis indicating a strong association with secondary metabolite biosynthetic processes. Temporal expression clustering revealed that only genes in Cluster 9 exhibited significant correlation with secondary metabolite production.

WGCNA (weighted gene co-expression network analysis) further pinpointed the green module as exhibiting the highest correlation with vanillin biosynthesis. Within this module, we identified 10 hub genes, including *HPP92_012558*, an NAC family transcription factor. The intersection of these three gene sets (common upregulated genes, Cluster 9, and the WGCNA green module) yielded 433 high-confidence candidates, among which 23 transcription factors and 18 functional proteins were directly linked to vanillin biosynthesis.

Our analysis of the top 10 most strongly differentially expressed transcription factors identified key regulators from multiple families, with NAC transcription factors emerging as the most prominent TFs, including *NAC-2* (*HPP92_014056*) and *NAC-3* (*HPP92_012558*), confirming their central role in vanillin biosynthesis regulation. These findings not only advance our understanding of secondary metabolite synthesis in vanilla orchids but also identify potential targets for enhancing vanillin production through genetic and metabolic engineering.

## Figures and Tables

**Figure 1 plants-14-01922-f001:**
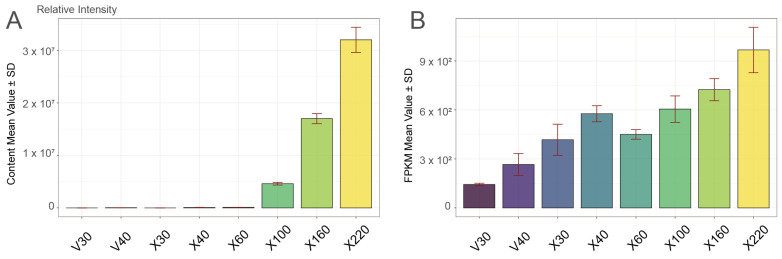
Metabolic and transcriptional profiling during vanilla (*Vanilla* spp.) growth and development. (**A**) Fluctuations in glucovanillin accumulation patterns across developmental stages. (**B**) Expression profiles of vanillin synthase gene *VpVAN* (*HPP92_026221*), with distinct expression patterns observed between low-vanillin *V. imperialis* (V) and high-vanillin *V. planifolia* (X) accessions.

**Figure 2 plants-14-01922-f002:**
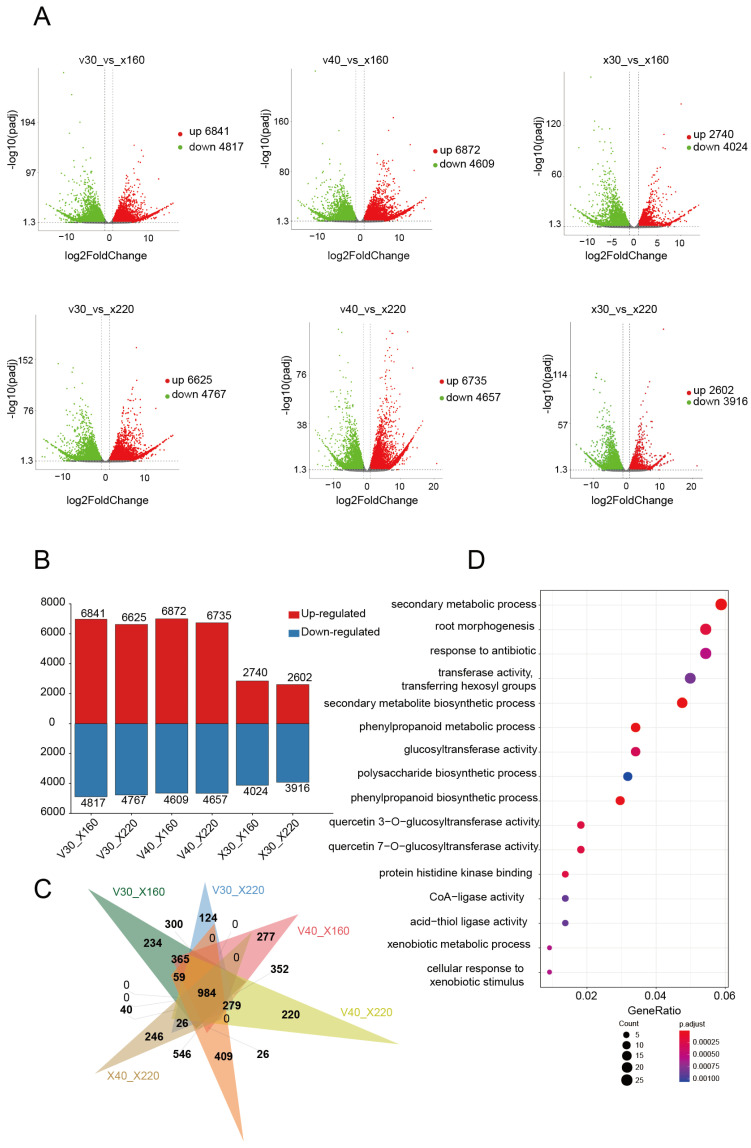
Differential gene identification and analysis. (**A**) Differentially expressed genes were identified among six pairwise comparisons and overall (**B**). (**C**) Venn diagram illustrates overlaps of upregulated differentially expressed genes across six comparison groups. (**D**) GO enrichment analysis was performed on 984 commonly upregulated differentially expressed genes. V: low-vanillin *V. imperialis*, X: high-vanillin *V. planifolia*.

**Figure 3 plants-14-01922-f003:**
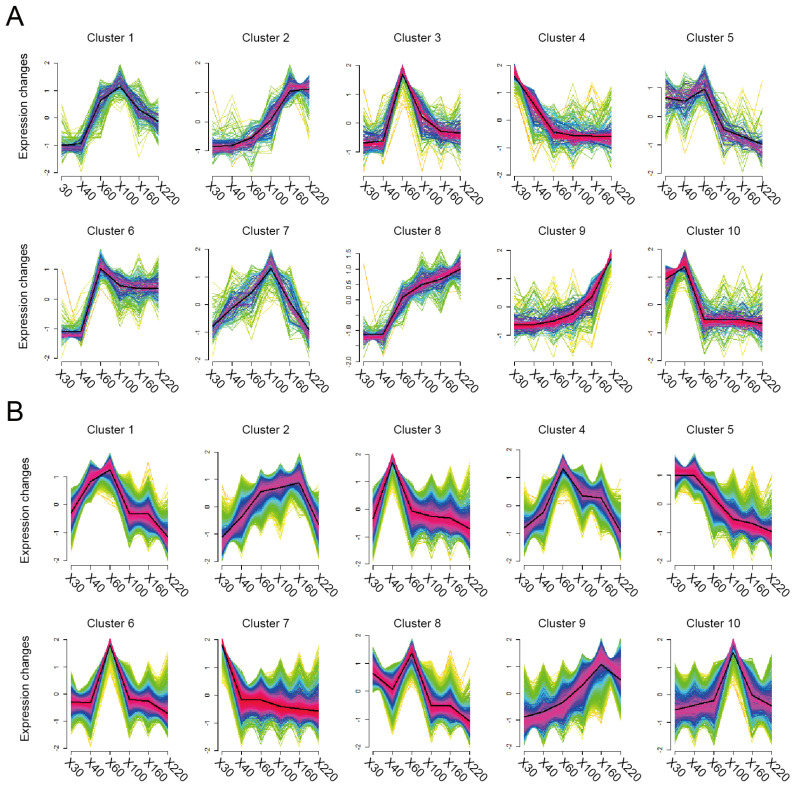
Expression patterns of metabolome (**A**) and transcriptome (**B**) in vanilla fruits and across six developmental stages, with different groups representing distinct expression trends; dark thick lines represent average expression profiles of all genes in each cluster.

**Figure 4 plants-14-01922-f004:**
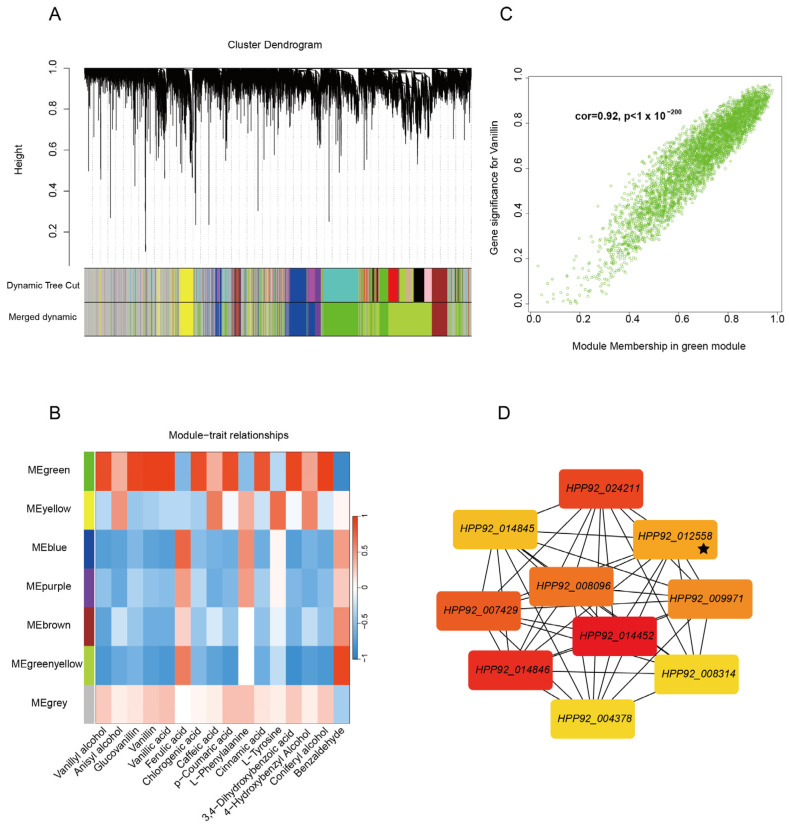
Co-expression network analysis of vanillin biosynthesis genes. (**A**) Hierarchical clustering trees with different topological overlaps of expressed genes. (**B**) Heat maps of correlations between modules and metabolites associated with vanillin synthesis. Different colors represent different modules (7 in total); numbers in grid represent Pearson correlations between modules and traits, ranging from −1 (blue) to 1 (red). Color gradient indicates strength of correlations, with darker blue signifying stronger negative relationships and deeper red representing more significant positive associations. (**C**) Relationship between gene significance (GS) and gene and module membership (MM) in MEgreen module. (**D**) Top 10 genes with highest Maximal Clique Centrality (MCC) scores were identified as hub genes. Black pentagrams indicate genes encoding NAC transcription factors.

**Figure 5 plants-14-01922-f005:**
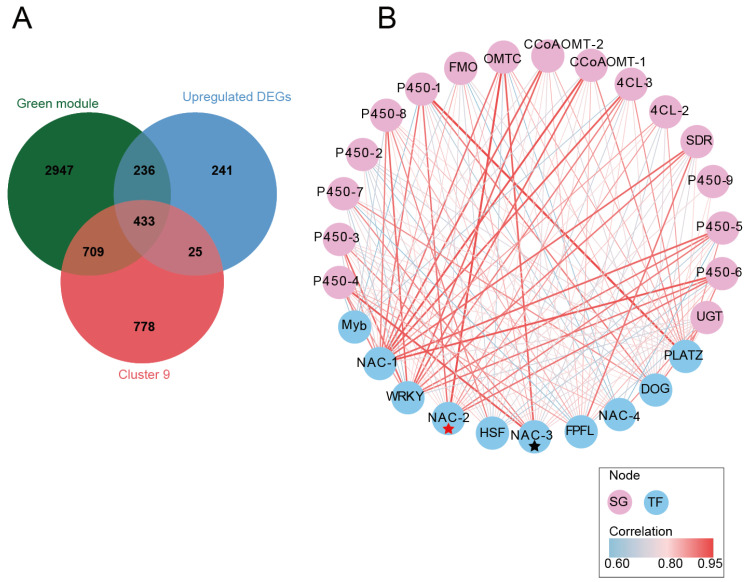
Integrated analysis. (**A**) The Venn diagram displays the overlapping upregulated differentially expressed genes among six comparison groups, Cluster 9, and the vanillin-related WGCNA module. (**B**) The red pentagrams denote the highest-scoring hub genes in the co-expression network, while the black pentagrams indicate the second-highest-scoring hub genes, one of which also emerged as the top-scoring hub gene in the WGCNA. Only Pearson’s r ≥ 0.6 are displayed.

## Data Availability

The original contributions presented in the study are included in the article/Appendix A; further inquiries can be directed to the corresponding author. The raw RNA-seq data generated from this study have been submitted to NCBI SRA (https://www.ncbi.nlm.nih.gov/sra, accessed on 20 May 2025) under the accession number PRJNA1264038.

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
