# Peer review of "Time-Series Metabolome and Transcriptome Analyses Reveal the Genetic Basis of Vanillin Biosynthesis in Vanilla"

_plants, 2025, doi:10.3390/plants14131922_

Round 1
Reviewer 1 Report
Comments and Suggestions for Authors
1.6.2025
The manuscript entitled "Time-Series Metabolome and Transcriptome Analyses Reveal the Genetic Basis of Vanillin Biosynthesis in Vanilla” was reviewed.
The manuscript delivers novel DEGs and metabolome data related to vanilla biosynthesis with temporal view for two contrasting species. However, the discussion was too short and weak, it was missing the temporal expression aspect which is the core of the study. In addition, qPCR validation of RNAseq data is missing.
Therefore, I do not recommend the publication of this manuscript in "Plants" unless these corrections are made. Please see below additional comments.
- General:
- The language is o.k. however, please check the entire manuscript again.
- The supplementary figures are very good; however, may I recommend moving Figure S3 (Volcano plots) to the main manuscript for its importance.
- Abstract:
- Line 20: replace “metabolomic,” with “metabolome analysis reveal”.
- Line 20: replace “temporal expression genes” with “genes with temporal expression”.
- Line 21: add “were” before “identified”.
- Line 21: replace “revealed six” with “revealing six major”.
- Line 22: replace “transcription factors” with “TFs” as you have already abbreviated in the preceding sentence.
- Lines 23-24: replace “transcription factor” with “TF”.
- Line 25: add “were” before “identified”.
- You still need to write all abbreviations in full the first time they appear in the abstract, e.g. DEGs, WGCNA, NAC, … etc. The only one you mentioned in full was TF!
- “Time-series analysis” is the major theme of your study and a major pillar in the title, however, you did not mention related findings in the abstract except for “2058 genes with temporal expression”, which is not enough. This is your novelty in the manuscript and need to be highlighted in more details in the abstract.
- Introduction:
- Good and informative.
- Lines 39-44: move this biosynthesis related sentences to the subsequent paragraph.
- Lines 45-49: move theses plant related sentences to the preceding paragraph.
- Lines 80-83: remove the sentences “By combining … biosynthesis” as you have already mentioned the aim of your study in details (Lines 73-80).
- Results:
- Well presented.
- Lines 86-91: delete these sentences as they reflect a research aim rather than results.
- Lines 99-104: delete these sentences as you are repeating yourself again. You do not need to repeat the materials and methods section.
- Figure 1: what is the unit for measuring unit for glucovanillin in panel A, you need to add it to the y-axis.
- Line 135: replace “regulatory” to “expression”.
- Figure 2: the Venn diagram in panel B with triangles is a big miss, you cannot figure out which is which. Please replace it with either circles or ovals!
- Lines 200-226: these two huge paragraphs are talking about CHILLING in okra (Abelmoschus esculentus)! However, you are studying vanilla!!! I think they were there by mistake, just remove them.
- Line 276: replace “clust” with “cluster”.
- Line 342: the “value in parentheses represents the P-value” is missing from panel B!
- Discussion:
- Short and weak, you have novel findings that need to be discussed including two major areas:
1) The temporal gene expression of major vanilla biosynthesis enzymes and their overlapping with novel TFs.
2) The huge variation in DEGs and TFs between the two investigated vanilla species.
- Line 429: both cited reference “53” and “54” are the same one! Remove “54” from the reference list.
- You need to discuss your finding in more details with related articles including “14” and “53”.
- Materials and Methods:
- Lines 457-463: you need to mention the followings:
1) Growing condition of vanilla plants under investigation, including fertilization, pest control, pollination, trellising and soil conditions.
2) Study season and year in addition to the harvest period.
3) Plant age.
- Line 475: remove space from “4 °C”.
- Line 477: remove space from “4 °C”.
- Lines 516-528: to validate any RNAseq data, qPCR need to be run for random DEGs, which is missing here, please you need to add this section.
- Conclusion:
- Lines 565-575: remove or cut short with modification as they are just a repetition to the abstract.
- References:
- The list is up-to-date (more than 50% of cited articles are published in the last five years.
- References “49” and “51” are the same, Wang et al 2022. Remove “54”.
- References “53” and “54” are the same, Gastelbondo et al 2025. Remove “54”.
Reviewer 2 Report
Comments and Suggestions for Authors
The authors conducted a comparative transcriptomic and metabolomic analysis of Vanilla species with high and low vanillin yields. They identified six TF classes associated with the regulation of vanillin biosyn genes. Two NAC transcription factors emerged as central regulators and may as promising targets for future metabolic engineering efforts in the vanilla industry.
The followings can improve the quality of the manuscript.
Abstract: Specify what “further comprehensive investigation” is required.
Line 39, 58: Provide a figure(s) for the synthetic pathway.
Fig S1B: Explain how to interpret the graph. What do the negative values mean? What are the data groups? CK and EG?
Line 99: What activities does vanillin, but not glucovanillin, play in the plant? Provide reference(s) for glucovanillin being inactive.
Line 103: Why only two data points for V. imperialis? Would comparing V120 vs X120 provide a simpler way to study vanillin biosynthesis given that other developmental processes are similar between the two samples? V30 and X220 are at different developmental stages (as stated in line 170) in addition to being different in vanillin production.
Line 101: Why did the authors analyze the dynamics of glucovanillin, rather than those of vanillin or the combined levels of vanillin and glucovanillin? Would it be the combined levels more accurately reflect the accumulation dynamics?
Line 118: The content increased from X160 to X220. It is not known if the content kept increasing after X220. How can the authors conclude that the level reached at maximum? What does ‘a metabolic steady state’ mean? The level at X100, for instance, is not at steady state?
Line 135: What distinct regulatory patterns were observed? Please be specific.
Line 162: Often, >10,000 DEGs were observed. How many genes were analyzed in a single comparison and how many non-DEGs were there?
Fig 2A: How to explain the large difference in the number of DEGs of V30_X160 and X30_X160? Same for V30_X220 and X30_X220?
Fig 2B: Please explain how ‘GeneRatio’ in the x-axis is calculated?
Line 200: What are Ae171 and Ae182? Why was chilling stress was tested? This experiment was not described in Methods and elsewhere. Could be a mix-up with a different manuscript (line 200 – 226).
Line 251, Fig 3B: Cluster9 showed a decrease from X160 to X220, whereas the metabolite Cluster2 showed little change from X160 to X220. It is not correct to say that the gene expression Cluster9 reached peak conc at 160-220 days (line 258).
Line 266: Explain how ‘responses to antibiotics’ and ‘monocarboxylic acid metabolic processes’ support vanillin biosynthesis.
Line 342: There is no value in parentheses in Fig 4B?
Lines 309-312: Explain how ‘response to metal ions,’ ‘cation binding’ and ‘monocarboxylic acid metabolic processes’ are connected to vanillin biosynthesis, particularly in terms of precursor supply and coenzyme coordination mechanisms.
Line 366: Provide a reference(s) for the roles of the SDR family members. Or, is this a speculation made by the authors?
Fig 5B: Why are there only 17 SGs, not 18, shown in Fig 5B? Table S7 has one more P450 SG.
Line 429: References 53 and 54 are the same.
Line 429: Compare in detail the findings in Reference 53 regarding NAC TFs with the findings in this study. Does this study confirm the earlier findings in Ref 53? Does this study report some findings that were not reported in Ref. 53?
Minor suggestions:
In Abstract, provide the full description of WGCNA.
Line 276: cluster
Lines 269 – 272: Rewrite.
Throughout the text, use hyphens, minus signs, and dashes clearly and consistently. Foe examples, (minus)80 in line 467 and 0(dash)1.5 min in line 485.
Lines 533, 539: MEgreen
Round 2
Reviewer 2 Report
Comments and Suggestions for Authors
The authors have adequately addressed most of the previous suggestions.
A few further suggestions are provided below.
Figure S1: In the figure caption, include a detailed description of each pathway, specifying its origin (e.g., plant, microbial, or synthetic), the name of the enzyme associated with each step, and relevant reference(s).
Figure S2B: It would help if a different symbol (e.g. square) is used for CK data points. Also, another symbol of ‘QC’. Currently, it is difficult to distinguish the CK and EG data, e.g., CK-P40 and EG-80 points. Also, include an explanation for the percentage values. You may include an interpretation of the graph like “the significant separation between CK and EG along PC1 (explaining 72.3% variance) confirms metabolic profile differences, validating the detection precision.”
Experimental design: This reviewer is not convinced that comparing V30 vs X220 allowed ‘clear functional genomic insights into differential production’, when ‘developmental differences exist between V30 and X220.’ Among the DEGs identified, some of them must have been differentially expressed due to developmental differences. This is especially so when considering the large numbers of DEGs identified, 11658, 11392, etc. The authors should address this concern and summarize why this concern does not invalidate this study.
Number of genes: The total number of genes, 29,168, should be given in the ms (L161) as a reference number for the different numbers of DEGs.
Fig 2A: The question was why there are much more DEGs in V30_X160 compared to X30_X160. This suggests that there are other differences in the two species in addition to vanillin biosynthesis. The authors suggested that ‘significant transcriptional divergence in vanillin synthesis pathways compared to its peak biosynthesis period’ in V. imperiallis. It is not clear what ‘transcriptional divergence’ means. Different regulatory pathways involving different transcription factors? It is hard to expect that different TFs are involved in vanillin biosynthesis depending on the level of biosynthesis. This should be addressed in a clear manner in the text or in the figure caption.
Fig 2C: The Venn diagrams are mis-labeled. For instance, the top left shows V30_X160 and V30_X220.
L284: Full description of CAR
L285: This reviewer was unable to find any mention of CAR stabilization by metal ions in reference [30]. Provide specifics (which metal ions, etc.) reported in reference [30].
L460: COMT
L461: Provide reference numbers of ‘previous reports.’
